# Safety of Bedside Placement of Tunneled Hemodialysis Catheters in the Intensive Care Unit: Translating from the COVID-19 Experience

**DOI:** 10.3390/jcm10245766

**Published:** 2021-12-09

**Authors:** Mohammad Ahsan Sohail, Tarik Hanane, James Lane, Tushar J. Vachharajani

**Affiliations:** 1Department of Internal Medicine, Cleveland Clinic Foundation, Cleveland, OH 44195, USA; 2Department of Critical Care Medicine, Respiratory Institute, Cleveland Clinic Foundation, Cleveland, OH 44195, USA; hananet@ccf.org (T.H.); lanej3@ccf.org (J.L.); 3Cleveland Clinic Lerner College of Medicine, Case Western Reserve University, Cleveland, OH 44195, USA; vachhat@ccf.org; 4Department of Nephrology and Hypertension, Glickman Urological & Kidney Institute, Cleveland Clinic Foundation, Cleveland, OH 44195, USA

**Keywords:** COVID-19, tunneled dialysis catheter, intensive care unit, anatomic landmarks, bedside procedure, ultrasound

## Abstract

Background: Critically ill patients with coronavirus disease 2019 (COVID-19) and kidney dysfunction often require tunneled hemodialysis catheter (TDC) placement for kidney replacement therapy, typically under fluoroscopic guidance to minimize catheter-related complications. This entails transportation of patients outside the intensive care unit to a fluoroscopy suite, which may potentially expose many healthcare providers to COVID-19. One potential strategy to mitigate the risk of viral transmission is to insert TDCs at the bedside, using ultrasound and anatomic landmarks only, without fluoroscopic guidance. Methods: We reviewed all COVID-19 patients in the intensive care unit who underwent right internal jugular TDC insertion at the bedside between April and December 2020. Outcomes included catheter placement-related complications such as post-procedural bleeding, air embolism, dysrhythmias, pneumothorax/hemothorax, and catheter tip malposition. TDC insertion was considered successful if the catheter was able to achieve blood flow sufficient to perform either a single intermittent or 24 h of continuous hemodialysis treatment. Results: We report a retrospective, single-center case series of 25 patients with COVID-19 who had right internal jugular TDCs placed at the bedside, 10 of whom underwent simultaneous insertion of small-bore right internal jugular tunneled central venous catheters for infusion. Continuous veno-venous hemodialysis was utilized for kidney replacement therapy in all patients, and a median catheter blood flow rate of 200 mL/min (IQR: 200–200) was achieved without any deviation from the dialysis prescription. No catheter insertion-related complications were observed, and none of the catheter tips were malpositioned. Conclusions: Bedside right internal jugular TDC placement in COVID-19 patients, using ultrasound and anatomic landmarks without fluoroscopic guidance, may potentially reduce the risk of COVID-19 transmission among healthcare workers without compromising patient safety or catheter function. Concomitant insertion of tunneled central venous catheters in the right internal jugular vein for infusion may also be safely accomplished and further help limit personnel exposure to COVID-19.

## 1. Introduction

The global coronavirus disease 2019 (COVID-19) pandemic, caused by the novel severe acute respiratory syndrome coronavirus 2 (SARS-CoV-2), has posed unique challenges for infection prevention and control within healthcare facilities [1]. Critical care providers, in particular, are currently faced with multiple challenges pertaining to the clinical management of patients with COVID-19 such as preventing infection transmission, maintaining a safe work environment for the intensive care unit (ICU) staff, and minimizing the stress on workforce capacity and infrastructure [2].

Kidney involvement in critically ill patients with COVID-19 is commonly observed [3], often requiring placement of a hemodialysis catheter to initiate kidney replacement therapy [4,5]. However, most patients require kidney replacement therapy for more than one week, necessitating transition from a non-tunneled dialysis catheter to a tunneled dialysis catheter (TDC). The standard clinical practice for TDC placement involves using ultrasound and fluoroscopic guidance, ensuring minimal immediate procedural complications with appropriate catheter tip positioning in the mid-right atrium [6]. This entails transporting the patient from the ICU to a fluoroscopy suite and potentially exposing numerous healthcare professionals and ancillary staff to SARS-CoV-2-infected patients [2]. One potential strategy to limit the exposure of multiple hospital staff members to COVID-19 involves bedside placement of TDCs in the ICU under continuous cardiac monitoring, using anatomic landmarks and ultrasound only, without fluoroscopic guidance [7].

However, it should also be acknowledged that the use of fluoroscopy reduces the risk of catheter placement-related complications [6], which contribute significantly to patients’ morbidity and mortality as well as increased economic burden to society [8]. Consequently, bedside TDC placement using anatomic landmarks should only be performed in well-selected patients to ensure that we do not compromise patient safety or quality of care in an attempt to reduce COVID-19 transmission among healthcare personnel.

In this study, our objective was to evaluate the safety and effectiveness of bedside TDC placement without fluoroscopic guidance in carefully selected patients with COVID-19. We hypothesized that bedside TDC placement in select patients by experienced providers in the ICU utilizing ultrasound guidance and anatomic landmarks, without having to transport SARS-CoV-2-infected patients to the fluoroscopy suite, can be successfully accomplished without compromising patient safety or catheter function.

## 2. Materials and Methods

### 2.1. Study Design, Location, and Outcomes

We report a retrospective, single-center case series conducted at the Cleveland Clinic in Cleveland, Ohio. The study protocol was reviewed and approved by the Institutional Review Board at the Cleveland Clinic, and the protocol was exempted from the requirement for informed consent from patients since only de-identified data collected during hospital visits were included in the study.

The primary objectives of this study were to evaluate the safety and efficacy of bedside TDC placement in patients with COVID-19 in the medical ICU without fluoroscopic guidance, specifically investigating the incidence of immediate procedure-related complications and catheter dysfunction following TDC insertion. Outcomes considered to evaluate the safety of bedside TDC insertion included incidence of procedural complications such as bleeding, arterial puncture, venous air embolism, arrhythmias, pneumothorax, hemothorax, and catheter tip malposition [9,10]. The optimal position for the tip of the TDCs was considered to be either within the mid-right atrium or at the junction of the superior vena cava (SVC) and the right atrium as confirmed with a post-procedure chest radiograph [11].

Outcomes considered to assess the efficacy of bedside TDC placement included dialysis prescription parameters such as blood flow rate and ultrafiltration rate. TDC insertion was considered to be successful when the catheter was able to sustain the prescribed blood flow to perform either a single intermittent or 24 h of continuous hemodialysis treatment.

### 2.2. Study Population and Selection Criteria for TDC Insertion

We conducted a retrospective chart review of all adult patients with COVID-19 in the medical ICU who underwent right internal jugular (IJ) TDC placement at the bedside utilizing anatomic landmarks and ultrasound guidance between 1 April 2020 and 31 December 2020 at the Cleveland Clinic. The diagnosis of COVID-19 infection was confirmed with a positive nucleic acid amplification test for SARS-CoV-2. Pre-procedural inclusion and exclusion criteria for patients prior to bedside TDC insertion are delineated in Table 1.

### 2.3. Protocol for Bedside Tunneled Dialysis Catheter Placement

Consultations for tunneled dialysis access from the medical ICU in COVID-19 patients with prior initial placement of non-tunneled hemodialysis catheters were reviewed carefully by the ICU procedural team to determine eligibility for bedside TDC insertion based on the patient selection criteria mentioned in Table 1. After identification of an appropriate candidate, the medical ICU procedure room was notified, and a cart with the necessary equipment and multiple lengths of TDCs was prepared.

The procedure was performed at the bedside using personal protective equipment (PPE) for COVID-19 as per hospital policy and standard aseptic practices [12,13] by a trained attending proceduralist, experienced with TDC placements under fluoroscopy, assisted by one procedure nurse. The attending proceduralists included one interventional nephrologist and two medical intensive care physicians. The ICU physicians were competent in placing non-tunneled dialysis catheters/CVCs and were trained to insert bedside TDCs by the experienced interventional nephrologist. They underwent a structured training program, involving simulation-based skills development, followed by observing two procedures, assisting with three, and independently performing ten procedures under close supervision. In addition, they completed online courses on fluoroscopy and radiation safety, which allowed them to receive hospital privileges. After independently completing 25 TDC insertions under fluoroscopy, the ICU physicians were eligible to apply for a relevant certification, issued by the American Society of Diagnostic and Interventional Nephrology. Sedation for the procedure was managed by the attending physician and the patient’s bedside ICU nurse. The procedure room was situated within the medical ICU, obviating the need to transport patients outside the confines of the medical ICU to one of the interventional radiology fluoroscopy suites located in the basement. TDC placement into the right IJ vein was considered to be a safe initial protocol for bedside insertion of TDCs under continuous cardiac monitoring without fluoroscopic guidance.

The protocol for right IJ TDC insertion using anatomic landmarks as previously published [14] can be briefly described as using the manubrial-sternal angle as the topographical landmark that corresponds to the carina, with the insertion depth estimated by measuring the distance between the skin venipuncture site 1–2 cm above the clavicle and a point 5 cm below the manubrial-sternal angle, and the catheter length determined by adding the insertion depth to the tunnel length (Figure 1 and Figure 2).

With the patient in supine position in their ICU bed, the right side of the neck and chest was prepared using chlorhexidine prep and sterile drapes. The right IJ vein was identified using a sterile ultrasound probe, and 2% lidocaine was injected at the puncture site. Using a micro-puncture set (21G needle, 4F coaxial sheath, and 0.018” wire), the right IJ vein was punctured under ultrasound guidance with subsequent advancement of a 0.018” wire. The coaxial sheath was withdrawn, and the 0.018” wire was replaced with a 0.035” wire under continuous cardiac monitoring to evaluate for cardiac ectopy, providing indirect evidence of the location of the wire tip in the right atrium. Lidocaine was infiltrated at the chosen exit site, and a small skin incision was made with the tunneled catheter advanced into the tunnel. The Bard Equistream™ (BD Inc., Franklin Lakes, NJ, USA), which is a double lumen TDC with a split tip design, was utilized for all bedside TDC insertions.

Using Seldinger’s technique, the venotomy was serially dilated, and a peel-away sheath was advanced and the wire withdrawn. The catheter was advanced by gradually peeling off the sheath. The threshold to abandon the procedure was low if there was any difficulty advancing the wire or the dilators. All catheter ports were checked for adequate flow with a 10 mL syringe, flushed and locked with 4% citrate solution, and sterile dressing was applied. The catheter tip position was confirmed with a post-procedure portable chest radiograph (Appendix A).

### 2.4. Data Acquisition and Statistical Analyses

The Cleveland Clinic has utilized an electronic health record (EHR) for inpatient and outpatient care since 2001. Data gathered from the EHR included age at the time of TDC insertion, sex, race/ethnicity, comorbidities, body mass index, reason for ICU admission, indication for TDC placement, use of antiplatelet/anticoagulant medications, the Acute Physiology and Chronic Health Evaluation (APACHE II) score, catheter placement-related complications, and components of dialysis prescription.

Race/ethnicity was identified based on self-description as recorded in the EHR, and was categorized as Caucasian, African American, Hispanic, Asian, or other (categorized as other when they were not classified as any of the aforementioned ethnicities or when race data were not available). Comorbid conditions included hypertension, diabetes mellitus, dyslipidemia, malignancy, coronary artery disease, cerebrovascular disease, end-stage liver disease, chronic kidney disease, and congestive heart failure. 

Catheter placement-related complications included bleeding, inadvertent arterial puncture, venous air embolism, arrythmias, pneumothorax, hemothorax, and catheter malposition [9,10]. Procedure-related clinically significant bleeding was defined as any bleeding requiring blood transfusions or any additional and unexpected hemostatic measures [15]. The desired position for the catheter tip was considered to be in the mid-right atrium or at the junction of the SVC and the right atrium, and catheter malposition was defined as either incorrect initial positioning of the catheter tip, displacement of the catheter tip after placement, or folding/kinking of the catheter tip [11]. The complications observed in this study were classified according to the reporting standards of the American Society of Diagnostic and Interventional Nephrology (ASDIN) [16].

The TDC placement procedure was classified as successful if the catheter was able to achieve blood flow and circuit venous pressures sufficient to perform a single intermittent hemodialysis treatment or a 24-h treatment with continuous kidney replacement therapy without significantly changing the dialysis prescription. The components of the dialysis prescription included the choice of hemodialysis membrane, dialysis session length, dialysate composition and temperature, blood flow rate, amount and rate of ultrafiltration, and method of anticoagulation.

Descriptive statistics of patient characteristics were presented in median and interquartile range for continuous variables, and in frequencies and percentages for categorical variables.

## 3. Results

### 3.1. Baseline Patient Characteristics

A total of 25 patients infected with COVID-19 who underwent right IJ TDC placement at the bedside in the ICU between 1 April and 31 December 2020, utilizing only anatomic landmarks and ultrasound for real-time guidance without fluoroscopy, were included in the analysis. Twenty-one patients had clinically apparent COVID-19 pneumonia at the time of TDC placement, requiring varying degrees of respiratory support (mechanical ventilation (MV) (*n* = 11); non-invasive ventilation (NIV) (*n* = 5); high-flow nasal oxygen (HFNO) (*n* = 2), and nasal cannula (*n* = 3)); four COVID-19-infected patients did not exhibit any acute respiratory illness and were admitted to the ICU for other indications including management of gastrointestinal bleeding (*n* = 2) and hepatorenal syndrome (*n* = 2). Tunneled central venous catheters (CVCs) (Bard Powerline™ [BD Inc., Franklin Lakes, NJ, USA]; small bore (5 Fr); single or double lumen) for infusion were concomitantly inserted into the right IJ vein in 10 of these patients. The patients in our cohort had a median age of 62 years (interquartile range (IQR): 55–70), including 18 men (72%) and 16 African Americans (64%), with a median body mass index of 28.8 (IQR: 25.2–33.2) kg/m^2^ and median APACHE II score of 22 (IQR: 19–25). 

Baseline kidney function was available for all our patients, 11 of whom did not have a history of chronic kidney disease with a baseline estimated glomerular filtration rate (eGFR) of greater than or equal to 60 mL/min/1.73 m^2^. Eight and five patients had chronic kidney disease stages 3 (eGFR of 30–59 mL/min/1.73 m^2^) and 4 (eGFR of 15–29 mL/min/1.73 m^2^), respectively. Only one of our patients had been on hemodialysis prior to hospitalization via a right upper extremity arteriovenous fistula and required TDC insertion after initial placement of a non-tunneled hemodialysis catheter, for continuous renal replacement therapy (CRRT). For all other patients in our series, the indication for replacement of initial non-tunneled hemodialysis access with a TDC was acute kidney injury anticipated to require hemodialysis support for longer than 1 week. Other comorbid conditions included diabetes mellitus (*n* = 12), hypertension (*n* = 18), congestive heart failure (*n* = 7), and end-stage liver disease (*n* = 2).

### 3.2. Catheter Function Following Bedside TDC Placement

Patients in our series had their initial non-tunneled hemodialysis access replaced through a fresh venipuncture with a TDC at bedside a median of 6 days (IQR: 4–7) after initiation of hemodialysis. Continuous veno-venous hemodialysis (CVVHD) was the CRRT modality employed in all patients. Median catheter blood flow, dialysate flow, and ultrafiltration rates of 200 (IQR: 200–200) mL/min, 2000 (IQR: 1800–2300) mL/h, and 100 (IQR: 100–200) mL/h were achieved, respectively, without any deviations from the dialysis prescription.

### 3.3. Safety Outcomes Following Bedside TDC Placement

No catheter placement-related complications including bleeding, arterial puncture, venous air embolism, arrhythmias, pneumothorax, or hemothorax were observed in our patients for the duration of their hospitalization. None of the catheter tips were malpositioned on post-insertion chest radiographs, with the tip located either in the mid-right atrium (*n* = 22) or at the junction of the SVC and the right atrium (*n* = 3).

## 4. Discussion

Medical ICU practitioners and hospital administrators may prepare for a potential surge in critical care capacity during the global COVID-19 pandemic by optimizing workflows for rapid diagnosis, isolation, and clinical management and by addressing concerns relating to supplies and infrastructure [17]. However, these preparations must also focus on the ICU staff, including protection from nosocomial transmission [18]. This is particularly crucial with hypoxic COVID-19 patients requiring non-invasive respiratory support, including NIV and HFNO, since these may generate aerosols composed of small virus-containing particles, depending on factors such as duration of use, oxygen flow velocity, mask leakage, and patient coughing and cooperation [19]. Our study was conceptually innovative in attempting to explore bedside TDC insertion as a viable alternative in order to help prevent hospital-acquired infections without compromising patient safety or quality of care. We describe a series of 25 patients infected with COVID-19 requiring kidney replacement therapy that underwent bedside placement of right IJ TDCs in the ICU, using only ultrasound and anatomic landmarks without fluoroscopic guidance.

The standard of care for TDC insertion dictates that the procedure be performed under ultrasound and fluoroscopic guidance to ensure fewer catheter placement-related complications and adequate catheter function [6]. Complications arising from TDC insertion, including catheter-related infections and thrombosis, contribute significantly to patients’ morbidity and mortality at a considerable economic burden to society [8,20]. The hypercoagulable state of patients with COVID-19 also makes them potentially more vulnerable to thrombotic complications and catheter occlusion [21,22]. Furthermore, it is important to ensure that the patients are comfortable whilst they are placed in the supine position in order to improve cooperation during the procedure and reduce the risk of insertion failure or catheter malpositioning [23]. However, it may be challenging to provide a balanced analgesic strategy during TDC placement that provides adequate discomfort relief whilst ensuring respiratory and cardiovascular stability, especially in patients on respiratory support requiring conscious sedation. Therefore, we implemented certain precautionary measures into our protocol for bedside TDC placement in patients with COVID-19 to mitigate the risk of developing catheter-related complications.

Firstly, we developed rigorous selection criteria for patients with COVID-19 under consideration for TDC insertion in an attempt to identify ideal candidates. In addition to excluding patients with absolute contraindications to TDC insertion, including sepsis/bacteremia and uncontrolled coagulopathies [6], we also did not attempt bedside TDC placement in patients with new-onset cardiorespiratory instability or with a history of central vein stenotic/occlusive disease.

Secondly, we selected the right IJ vein as the preferred site for bedside TDC placement to further try to minimize the risk of catheter dysfunction and placement-related complications. Blood flow rates are consistently higher with right-sided jugular venous catheters than with left-sided catheters since placement of a catheter into the left IJ vein requires that the catheter make two right-angled bends as well as an antero-posterior bend over the pulmonary arch prior to reaching the SVC [24]. A retrospective analysis of jugular venous catheters depicted that left-sided IJ venous catheters had higher rates of infection (0.50 vs. 0.27; *p* = 0.005) and catheter dysfunction (0.25 vs. 0.11; *p* = 0.036) compared with those inserted from the right [25]. Therefore, the longer path that a left-sided catheter has to traverse predisposes it to developing catheter dysfunction as opposed to a right-sided catheter, which has a relatively shorter, less meandering course to the SVC.

Utilizing our protocol and implementing the aforementioned safety measures, we were successfully able to place TDCs at the bedside in patients with COVID-19 requiring kidney replacement therapy and achieved adequate catheter blood flow rates without any deviation from the CVVHD prescription or evidence of catheter occlusion/dysfunction. Furthermore, no catheter placement-related complications were observed, and none of the catheter tips were malpositioned on post-insertion chest radiographs. The acute escalation in the proportion of critically ill COVID-19 patients with acute kidney injury and the overwhelming need for urgent kidney replacement resources, coupled with the prospect of having to transport all these patients outside the ICU to the fluoroscopy suite for TDC placement, mandated the consideration of bedside TDC insertion. In addition to protecting hospital personnel from COVID-19 exposure by limiting transportation needs, we further limited potential nosocomial transmission among healthcare workers by eliminating the involvement of multiple teams for the procedure.

Moreover, we were able to concomitantly place tunneled CVCs for infusion (small bore (5 Fr); single or double lumen) along with TDCs in the right IJ vein in 10 of our patients, potentially further reducing personnel exposure to COVID-19. One study comparing patients who underwent concomitant placement of a TDC and a CVC for infusion with patients who only had a TDC placed in their right IJ vein found no significant differences between the two groups of patients when evaluating incidence of thrombosis (1.0% vs. 0.0%, *p* > 0.999), line infection (2.1% vs. 0.0%, *p* = 0.519), or line dysfunction (2.1% vs. 0.0%, *p* = 0.516) [26]. No puncture-related complications such as pneumothorax were reported for either group. Therefore, although multiple central catheters are not routinely placed in the same vein, it can be considered in certain clinical situations without increasing the risk of complications.

It is encouraging to note that another case series at a tertiary community-based medical center has also reported successful placement of 24 bedside TDCs by the institution’s vascular surgery department [27]. Only one patient out of the series developed a catheter placement-related complication, which was pneumothorax and cardiac tamponade. Some important distinctions between our studies include: (a) Some of their TDC insertions utilized fluoroscopic guidance requiring movement of the patient to a fluoroscopy-compatible bed, whereas only ultrasound was used in all our procedures for real-time guidance; (b) Chest radiography was utilized at the time of sheath insertion in their series with a median of 5 plain radiographs taken during TDC insertion, whilst only one post-procedure chest radiograph was performed in our patients for confirmation of catheter tip position; (c) We have clearly stated the selection criteria for patient candidacy for bedside TDC insertion to identify ideal candidates in an attempt to mitigate the risk of catheter-related complications; (d) Concomitant placement of TDCs and tunneled CVCs in the right IJ vein was performed in our series.

## 5. Conclusions

In conclusion, preferential right IJ TDC placement at the bedside in a carefully selected group of patients with COVID-19, using ultrasound and anatomic landmarks without fluoroscopic guidance, may potentially reduce the risk of transmission of COVID-19 among healthcare workers without compromising patient safety or catheter function. Concomitant insertion of tunneled CVCs in the right IJ vein may also be safely accomplished and further help limit personnel exposure to COVID-19.

Lessons learned from our experience are valuable to improving the safety of TDC placement in regions where access to fluoroscopy may be limited. The standard teaching of utilizing both ultrasound and fluoroscopy guidance for TDC placement creates an impediment to providing timely care and dependence on other specialties [28]. Our experience with a carefully selected population may translate to several other clinical scenarios: (1) limited access to fluoroscopy in resource-limited countries; (2) regional and local regulatory restrictions on fluoroscopy use by physicians other than radiologists; (3) preventing transmission of infections including clostridium difficile, vancomycin-resistant enterococci, and other multidrug-resistant organisms; (4) patient-related limitations precluding the use of fluoroscopy (e.g., pregnancy); (5) challenges with transporting patients with difficult body habitus (e.g., morbid obesity).

## Figures and Tables

**Figure 1 jcm-10-05766-f001:**
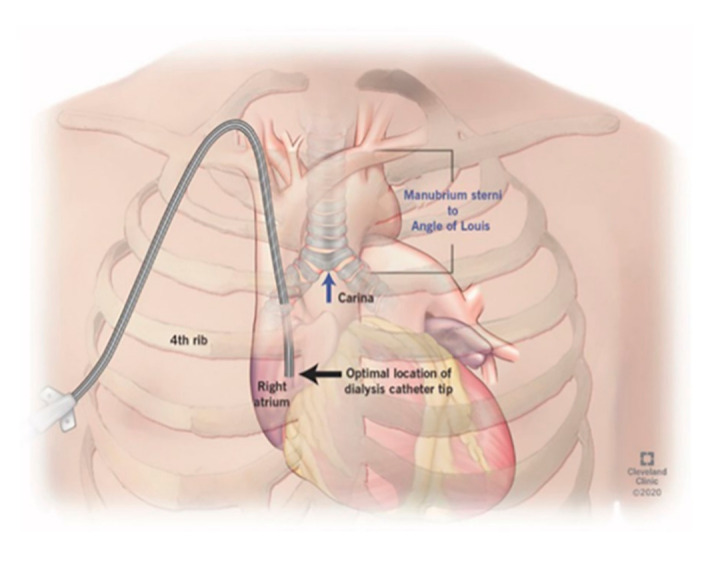
Surface anatomic landmarks for tunneled dialysis catheter placement: location of the carina in relation to the right atrium.

**Figure 2 jcm-10-05766-f002:**
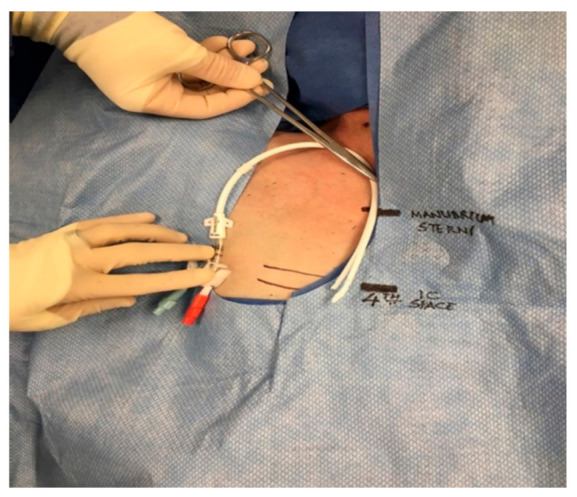
Using anatomic landmarks to estimate length of tunneled dialysis catheters.

**Table 1 jcm-10-05766-t001:** Pre-procedural inclusion and exclusion criteria prior to TDC insertion.

**Inclusion Criteria:**
**1.** Confirmation of patency of the right internal jugular vein via bedside ultrasound assessment
**2.** Appropriate withdrawal of anticoagulation (heparin and direct oral anticoagulants stopped 4 and 72 h prior to the procedure, respectively)
**3.** Arrangement of nothing by mouth status 4 h prior to procedure
**4.** Appropriate blood product transfusions to keep platelet count greater than 20,000 k/uL, INR less than 2, and normalize TEG parameters in patients with liver disease (reaction time less than 15 min, α-angle greater than 45 degrees, and maximum amplitude greater than 30 mm) to minimize risk of procedure-related bleeding complications ^1^
**Exclusion Criteria:**
**1.** History of thoracic central vein occlusive/stenotic disease
**2.** History of superior vena cava stent placement or recanalization
**3.** Hemodynamic instability defined as either a new requirement for vasoactive support or new-onset cardiac dysrhythmia within 4 h of scheduled TDC placement, or increase in existing vasoactive support of greater than 15 mcg of norepinephrine or greater than 150 mcg of phenylephrine within 2 h of planned TDC insertion
**4.** Respiratory instability defined as an inability to maintain an oxygen saturation of greater than 90% in the supine position, significant risk of aspiration in the supine position, or a recent increase in oxygen requirements defined as an increment in FiO2 of greater than 20% within 4 h of scheduled TDC placement
**5.** Blood cultures negative for less than 48 h prior to planned TDC insertion

TDC: tunneled hemodialysis catheter; INR: international normalized ratio; TEG: thromboelastography. ^1^ Kang, Y.G.; Martin, D.J.; Marquez, J.; Lewis, J.H.; Bontempo, F.A.; Shaw, B.W., Jr.; et al. Intraoperative changes in blood coagulation and thrombelastographic monitoring in liver transplantation. *Anesth Analg.*
**1985**, *64*(9), 888–896.

## Data Availability

All data generated or analyzed during this study are included in this published article.

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
