# Peer review of "Safety of Bedside Placement of Tunneled Hemodialysis Catheters in the Intensive Care Unit: Translating from the COVID-19 Experience"

_jcm, 2021, doi:10.3390/jcm10245766_

Round 1

Reviewer 1 Report

This an interesting article talking about the safety of placing tunnels dialysis catheter in COVID  patients.   I found it interesting and helpful. I have several questions and suggestions in the review of this paper.

  1. I am bit concerned  that the study was exempted from informed consent since the procedure was not standard of care and justification for need for a TDC rather the standard placement of US guided non tunneled catheter was not stated in the paper. The justification in the paper for THC uses reference 4,5 but neither of those articles state that tunneled catheter are preferred nor indicated in covid 19 patients.
  2. The article should state what the indications for a TDC were. Most ICU patients usually get a temporary inserted catheter for HD, until the decision is made that the patient will need some sort of chronic HD. . 10 of your patients were in the ICU for NC and HFLO resp support which i would think was unusual for patients to be in an ICU and certainly did not predict the need for long term HD. Of all your patient possible only 4 of them would be at high risk of long term HD. why exposed these people to a procedure that had higher risk than standard non-tunneled placement. Is it standard of care at your hospital to place tunnels lines in all patient needing emergent / temporary HD. If so that needs to be stated.
  3. Please state under location section what type of ICU these patients were located.
  4. You talk about an ICU procedural team. please describe. training and experience.  This would seemd to be a very important  factor as to the safety of the procedure 
  5. You often say in the paper that patient movement was a consideration for safety yet you moved all the patients to a procedural room.  Please explain the difference in safety
  6. What type of HD catheter was used.
  7. Please remove all tables, the description in the body are good enough
  8. The procedure is the most important  part of this paper Please add a diagram of the procedure
  9. The discussion is much to wordy  

Author Response

Reviewer # 1 Comments and Responses (Please see attachment)

This is an interesting article talking about the safety of placing tunneled dialysis catheter in COVID patients. I found it interesting and helpful. I have several questions and suggestions in the review of this paper.

Comment # 1: I am bit concerned that the study was exempted from informed consent since the procedure was not standard of care and justification for need for a TDC rather the standard placement of US guided non tunneled catheter was not stated in the paper. The justification in the paper for TDC uses reference 4,5 but neither of those articles state that tunneled catheters are preferred nor indicated in COVID-19 patients.

Response # 1: The study is a retrospective analysis of data collected on patients who received bedside TDC placement; hence the Institutional Review Board approved an exemption for obtaining an informed consent. However, the clinical decision to place a bedside TDC including the involved risks and benefits were explained to the consenting legal party as per standard clinical practice.

It was also mentioned in the ‘Results’ section under the subheading ‘Catheter Function Following Bedside TDC Placement’ that all patients in our series had a non-tunneled hemodialysis catheter placed initially and subsequently underwent TDC placement during the course of their hospitalization. This was articulated in the text as follows: “Patients in our series had their initial non-tunneled hemodialysis access replaced through a fresh venipuncture with a TDC at bedside a median of 6 days (IQR: 4-7) after initiation of hemodialysis.”

For further clarification, we can include the fact that all patients already had a non-tunneled dialysis catheter placed at the time of consideration of bedside TDC insertion in the ‘Materials and Methods’ section. This can be articulated in the text under the subheading ‘Protocol for Bedside Tunneled Dialysis Catheter Placement’ as follows: “Consultations for tunneled dialysis access from the medical ICU in COVID-19 patients with prior initial placement of non-tunneled hemodialysis catheters, were reviewed carefully by the ICU procedural team to determine eligibility for bedside TDC insertion based on the patient selection criteria mentioned in Table 1.”

Comment # 2: The article should state what the indications for a TDC were. Most ICU patients usually get a temporary inserted catheter for HD, until the decision is made that the patient will need some sort of chronic HD. 10 of your patients were in the ICU for NC and HFLO resp support which I would think was unusual for patients to be in an ICU and certainly did not predict the need for long term HD. Of all your patient possible only 4 of them would be at high risk of long-term HD. why exposed these people to a procedure that had higher risk than standard non-tunneled placement. Is it standard of care at your hospital to place tunneled lines in all patient needing emergent / temporary HD. If so that needs to be stated.

Response # 2: As mentioned in our response to comment # 1, all our patients did have non-tunneled dialysis catheters at the time of consideration of bedside TDC insertion. These patients had non-tunneled dialysis catheters for a median of 6 days (IQR:4-7) before TDCs were inserted through a fresh venipuncture. TDCs were indicated in these patients because acute kidney injury was anticipated to require hemodialysis support for longer than 1 week.

The indication for TDC placement can be delineated in the ‘Results’ section under the subheading ‘Baseline Patient Characteristics’ as follows: “Only one of our patients had been on hemodialysis prior to hospitalization via a right upper extremity arteriovenous fistula and required TDC insertion after initial placement of a non-tunneled hemodialysis catheter, for continuous renal replacement therapy (CRRT). For all other patients in our series, the indication for replacement of initial non-tunneled hemodialysis access with a TDC, was acute kidney injury anticipated to require hemodialysis support for longer than 1 week.”

Comment # 3: Please state under location section what type of ICU these patients were located.

Response # 3: The location of these patients was the medical ICU and this can be added in the location section as follows: “The primary objectives of this study were to evaluate the safety and efficacy of bedside TDC placement in patients with COVID-19 in the medical ICU without fluoroscopic guidance, specifically investigating the incidence of immediate procedure related com-plications and catheter dysfunction following TDC insertion.”

Comment # 4: You talk about an ICU procedural team. please describe. training and experience.  This would seem to be a very important factor as to the safety of the procedure

Response # 4:

The procedures were performed by one interventional nephrologist and two medical ICU physicians. The ICU physicians were competent in placing non-tunneled catheters and were trained to place TDC by the experienced interventional nephrologist. The ICU physicians received structured training, including simulation-based skills development followed by observing 2 procedures, assisting with 3 and independently performing 10 procedures under close supervision. The ICU physicians also completed an online course on fluoroscopy and radiation safety, which allowed them to receive hospital privileges. Additionally, after independently completing 25 procedures under fluoroscopy, the ICU physicians were eligible to apply for a certification from the American Society of Diagnostic and Interventional Nephrology Society (www.ASDIN.org).

Comment # 5: You often say in the paper that patient movement was a consideration for safety yet you moved all the patients to a procedural room.  Please explain the difference in safety

Response # 5: The procedure room in the medical ICU at our institution is located within the medical ICU. This procedure room was utilized for all bedside TDC insertions. Consequently, the patients only had to be transported within the confines of the medical ICU from their individual rooms to the medical ICU procedure room. If TDCs were placed under fluoroscopy, the patients would have to be transported to one of the interventional radiology fluoroscopy suites, which are located on a different floor in a separate building (the patient would have to be transported across hospital halls and elevators).

We can clarify in the ‘Materials and Methods’ section the exact location of the procedure room as follows: “The procedure room was situated within the medical ICU, obviating the need to transport patients outside the confines of the medical ICU, to one of the interventional radiology fluoroscopy suites located in the basement.”

Comment # 6: What type of HD catheter was used?

Response # 6: The Bard Equistream™ hemodialysis catheter was inserted in all our patients with a split tip design. We can add this information to the text under the subheading ‘Protocol for Bedside Tunneled Dialysis Catheter Placement’ as follows: “A double lumen tunneled catheter with a standard split tip design was utilized for all bedside TDC insertions.”

Comment # 7: Please remove all tables, the description in the body are good enough

Response # 7: We can remove Tables 2 and 3 from the text based on your suggestion. However, we have not removed Table 1 since the selection criteria utilized for assessing candidacy for bedside TDC insertion has not been listed anywhere in the text apart from Table 1.

Comment # 8: The procedure is the most important part of this paper Please add a diagram of the procedure

Response # 8:

Figure 1: Surface anatomical landmarks for tunneled dialysis catheter placement: location of the carina in relation to the right atrium.

Comment # 9: The discussion is much too wordy  

Response # 9: We have made an attempt to decrease the word count of the discussion section. We are open to any suggestions on how to reduce the content of the discussion but we believe that we have covered several important points in the discussion section:

  1. Discussed the rationale for attempting bedside TDC insertion: reducing COVID-19 transmission by obviating the need to transport patients outside the ICU and decreasing the number of medical teams involved in TDC insertion (eg. interventional radiology, anesthesia)

  1. Discussed the challenges faced in patients with COVID-19 including aerosol production with NIV and HFNO as well as potential for increased risk of complications including thrombotic complications

  1. Steps taken to reduce the risk of catheter related complications such as implementing rigorous selection criteria for bedside TDC insertion

4. Safety of concomitant placement of multiple tunneled catheters in the same central vein

Reviewer 2 Report

Thank you for this work - I read it with great interest. The study is methodologically sound and the results show potential for direct application. 

A few comments:

  1. This data may not be available, but since the goal for doing TDC at ICU bedside is in part to reduce potential transmission of COVID-19, were there any exposure events linked to TDC Placement?
  2. Did the same proceduralist perform each TDC placement? If so, what was their training (e.g., hospitalist, IR, radiology)? Your conclusion mentions physicians other than radiologists using fluoroscopy, and so it is useful to understand the training of the proceduralist for context.
  3. Were any exit site infections or other infectious complications observed during hospitalization for these patients?

Author Response

Reviewer # 2 Comments and Responses (Please see attachment)

Thank you for this work - I read it with great interest. The study is methodologically sound and the results show potential for direct application. 

A few comments:

Comment # 1

This data may not be available, but since the goal for doing TDC at ICU bedside is in part to reduce potential transmission of COVID-19, were there any exposure events linked to TDC Placement?

Response # 1: Although one of the objectives of performing bedside TDC placement is to ultimately reduce the risk of COVID-19 transmission amongst health care personnel, our retrospective case series analysis was focused on demonstrating the safety and efficacy of performing bedside TDC insertion using anatomic landmarks and ultrasound only, without fluoroscopic guidance. Consequently, data on exposure events linked to TDC insertion are not available to report.

Comment # 2

Did the same proceduralist perform each TDC placement? If so, what was their training (e.g., hospitalist, IR, radiology)? Your conclusion mentions physicians other than radiologists using fluoroscopy, and so it is useful to understand the training of the proceduralist for context.

Response # 2: The procedures were performed by one interventional nephrologist and two medical ICU physicians. The ICU physicians were competent in placing non-tunneled catheters and were trained to place TDC by the experienced interventional nephrologist. The ICU physicians received structured training, including simulation-based skills development followed by observing 2 procedures, assisting with 3 and independently performing 10 procedures under close supervision. The ICU physicians also completed an online course on fluoroscopy and radiation safety, which allowed them to receive hospital privileges. Additionally, after independently completing 25 procedures under fluoroscopy, the ICU physicians were eligible to apply for a certification from the American Society of Diagnostic and Interventional Nephrology Society (www.ASDIN.org).

Comment # 3

Were any exit site infections or other infectious complications observed during hospitalization for these patients?

Response # 3: Our retrospective case series study was focused on reporting immediate mechanical complications associated with catheter insertion such as pneumothorax, venous air embolism, arterial injury, arrhythmias and catheter malposition. Consequently, data on delayed complications (>1 week) such as catheter related infections as well as central vein stenosis or thrombosis are not available.

Round 2

Reviewer 1 Report

Please add in the protocol section the actual name of the catheter. 

Please reference a video showing the tunneled HD catheter technique

Author Response

Comment # 1: Please add in the protocol section the actual name of the catheter. 

Response # 1: We have mentioned the actual name of the TDC utilized in the protocol section as follows: “The Bard Equistream™, which is a double lumen TDC with a split tip design, was utilized for all bedside TDC insertions.”

We have also added information regarding the training and experience of the ICU procedural team, which was requested in the previous revision of the manuscript, to the procedure section as follows: “The attending proceduralists included one interventional nephrologist and two medical intensive care physicians. The ICU physicians were competent in placing non-tunneled CVCs and were trained to insert bedside TDCs by the experienced interventional nephrologist. The ICU physicians received structured training, involving simulation-based skills development, followed by observing two procedures, assisting with three and independently performing ten procedures under close supervision. In addition, they completed online courses on fluoroscopy and radiation safety, which allowed them to receive hospital privileges. After independently completing 25 procedures under fluoroscopy, the ICU physicians were eligible to apply for a certification from the American Society of Diagnostic and Interventional Nephrology.”

Comment # 2: Please reference a video showing the tunneled HD catheter technique.

Response # 2: We are in the process of preparing a video demonstrating the process of inserting a bedside tunneled dialysis catheter at our institution. We are requesting an additional one week’s time to be able to prepare and upload the video for inclusion into the manuscript as a web-link. Thank you for your understanding and consideration!

In the meantime, we have added another figure to the manuscript demonstrating the estimation of the length of tunneled dialysis catheters using surface anatomical landmarks.

See attached file for image. 
